# Effective Interaction between Quantization and Low-Rank Decomposition based on LLMs

## Abstract

As the parameter size of language models continues to grow, effective model compression is required to reduce their computational and memory overhead. Low-rank decomposition and quantization are two prominent compression methods that have been proven to significantly reduce the computational and memory requirements of Large Language Models (LLMs) while maintaining model accuracy. However, how these two methods interact when combined remains a critical question for developers, as many assume they are orthogonal, meaning their combination would not introduce additional errors beyond those independently introduced by each method. This paper provides the first mathematical proof that low-rank decomposition and quantization are non-orthogonal. We validate these findings through a series of experiments on large language models. Our results demonstrate that these methods are non-orthogonal, and their combination leads to significant performance degradation. Importantly, we propose a novel approach Diagonal Adhesive Method (DAM), which can effectively combine the two methods and mitigate the performance loss. Our research provides deep insights into model compression and lays a solid theoretical and experimental foundation for future related studies.

## 1 Introduction

In recent years, Large Language Models (LLMs) have demonstrated excellent performance in numerous natural language processing tasks, thanks to their increasing parameter counts (Huang et al., 2021; 2022). However, this growth in parameter size comes at the cost of significantly higher computational and storage demands (Meng et al., 2022). Consequently, efficient and low-cost deployment of LLMs has become a critical area of research (Gupta et al., 2023). Prior work in this field can be broadly categorized into two main approaches: Architecture-modifying techniques and Architecture-agnostic techniques (Ding et al., 2023).

Architecture-modifying techniques include distillation and pruning (Hwang et al., 2021). Distillation explicitly extracts knowledge from a large model and utilizes it to train a smaller one (Porada et al., 2021). Pruning, on the other hand, removes less important parameters to reduce computational and storage overhead. However, both distillation and pruning are often impractical for LLMs due to their substantial requirements for training data and compute resources. Architecture-agnostic techniques include quantization and low-rank decomposition (Levy et al., 2017). Quantization reduces the precision of model weights or activations, typically from 32-bit floating-point to lower bit representations like 8-bit or 4-bit integers, or even binary (Ashkboos et al., 2025). Low-rank decomposition approximates weight matrices with lower-rank matrices, reducing parameter count while keeping the weights in floating-point format (Yuan et al., 2023). Quantization and low-rank decomposition are popular choices for deploying LLMs practically because they are low-cost, architecture-agnostic, and generally perform well.

Existing quantization and low-rank decomposition methods both face performance bottlenecks (Sun et al., 2025; Yuan et al., 2023). For example, quantization can maintain good performance at precisions above W4A4KV4, but when the model is further compressed to lower bit-widths, there is a significant and unacceptable drop in accuracy (Hu et al., 2025). Similarly, low-rank decomposition suffers a notable decline in performance when the compression ratio exceeds 50% (Wang et al., 2024). Extensive experiments indicate that W4A4KV4 and a 50% compression ratio represent the

current bottlenecks for model compression techniques, limiting the potential for further high-fidelity compression. An inspiring idea is to effectively combine these two architecture-agnostic techniques.

It is natural to expect that directly combining quantization and low-rank decomposition can yield significant benefits in terms of computation and storage costs. However, the potential drawbacks remain unclear. Previous studies have assumed that these two methods are orthogonal, meaning their combination would not introduce additional errors beyond those of each individual method (Wang et al., 2024). However, these studies have primarily focused on quantizing weights only, making extra error from combining quantization and low-rank decomposition less noticeable, and overlooking the widespread presence of outliers when quantifying activation values. These studies have failed to properly investigate the overall impact of utilizing both methods together or to provide effective strategies for their integration. This poses a major challenge for further low-cost deployment of LLMs.

To address these challenges, we first conduct a theoretical analysis from both the tensor and dot-product perspectives, demonstrating that quantization and low-rank decomposition are non-orthogonal and thus introduce additional error. In addition, we find that the order in which these two methods are applied significantly affects model performance, and we derive the theoretically optimal sequence—applying low-rank decomposition before quantization. Finally, we identify outliers of activation as a key factor impacting compressed model performance, and propose a Diagonal Adhesive Method (DAM) that can significantly reduce the extra losses caused by the combination of quantization and low-rank decomposition.

To the best of our knowledge, we are the first to theoretically demonstrate that quantization and low-rank decomposition are non-orthogonal and provide the correct compression order. Based on the outliers issue inherent in low-rank decomposition, we propose the DAM method. Extensive experiments have shown that the DAM method significantly improves the performance of compressed models. Our research provides deep insights into model compression, and lays a solid theoretical and experimental foundation for future related studies. Our contributions are summarized below:

- We mathematically prove that quantization and low-rank decomposition are non-orthogonal operations. Based on compression error analysis, their combination introduces compound errors and leads to performance degradation. Our findings provide a theoretical foundation and challenge the conventional belief that combining quantization and low-rank decomposition does not significantly impact performance.

- To improve the performance of combining quantization and low-rank decomposition, we are the first to derive the optimal order—applying low-rank decomposition before quantization. This finding is further supported by extensive experimental results.

- We propose a Diagonal Adhesive Method (DAM) to address the performance degradation caused by the combination of quantization and low-rank decomposition. Extensive experiments demonstrate that while maintaining low cost and high speed, DAM significantly improves performance.

## 2 RELATED WORK

**Quantization.** Post-training quantization (PTQ) has emerged as a prominent technique for large language models (LLMs) due to its efficiency. Current PTQ methods can generally be categorized into weight-only and weight-activation quantization. To minimize memory usage, some strategies concentrate on weight-only quantization. GPTQ employs Hessian-based error compensation to achieve significant compression rates by reducing quantization errors (Frantar et al., 2022). AWQ (Lin et al., 2024) enhances performance by tackling the effects of activation outliers on weight quantization (Lee et al., 2023). QuIP (Chee et al., 2023) and QuIP# (Tseng et al., 2024) utilize random Hadamard matrices for incoherent processing and apply vector quantization to weights, resulting in improved performance compared to reduced precision quantization. SmoothQuant (Xiao et al., 2023) shifts the difficulty of quantization from activations to weights through a mathematical transformation. OmniQuant (Shao et al., 2023) further boosts performance by training quantization parameters and transformation coefficients. Additionally, I-LLM (Hu et al., 2024) proposes a strategy for integer-only quantization and inference through fully-smooth block reconstruction and fully integer operators. Recently, QuaRot (Ashkboos et al., 2025) employs random rotation matrices to

facilitate 4-bit quantization of weights and activations, while SpinQuant (Liu et al., 2024) learns these matrices to refine the 4-bit quantization process.

**Low-Rank Decomposition.** Singular Value Decomposition (SVD) is a common technique for reducing matrix size by approximating a matrix with two smaller low-rank matrices (Golub et al., 1987). In the realm of LLM compression, only a limited number of SVD-based methods have been suggested. Specifically, standard SVD focuses solely on compressing the original weight matrix without accounting for the significance of the parameters, which may result in a higher compression error. To tackle this issue, FWSVD method was introduced that incorporates Fisher information to assess parameter importance (Hsu et al.). However, this approach necessitates a complex gradient calculation, requiring significant resources for LLM compression. Another challenge with standard SVD is the distribution of activation, which can influence compression accuracy. To address this, ASVD method was proposed that scales the weight matrix utilizing a diagonal matrix to reflect the impact of input channels on the weights (Yuan et al., 2023). Nonetheless, both methods do not establish a clear relationship between singular values and compression loss.

**Combining Quantization and Low-rank Decomposition.** Previous studies have explored the combination of quantization and low-rank decomposition for model compression. SVD-LLM only compresses the model's weights without addressing the activation values and neglects their existence (Wang et al., 2024). Prior research has not provided clear conclusions regarding the optimal order of these two methods, nor has it offered solutions to the problem of outliers. These issues pose significant challenges for further high-quality model compression.

## 3 NON-ORTHOGONALITY OF QUANTIZATION AND LOW-RANK DECOMPOSITION

**Definition 3.1 (Quantization Method).** The existing quantization method is a block based quantization method that divides the weight matrix into multiple blocks and quantizes each block independently. For each block, utilize the maximum absolute value within that block as the scaling factor.

$$Q(W) = \text{Round}(\frac{W}{\max(|W|)} \cdot 2^{b-1}) \tag{1}$$

among them, $Q(W)$ represents the quantized block, $W$ represents the original weight block, $\max(|W|)$ represents the maximum absolute value of the elements in block $W$, $b$ represents the quantization bit width, Round($\cdot$) represents rounding operation. The inverse quantification formula is

$$D(Q(W)) = Q(W) * \max(|W|)/(2^b - 1) \tag{2}$$

**Definition 3.2 (Quantization Error).** We formalize the quantification error and theoretical error boundary as follows

$$E(Wx) = |Wx - D(Q(W))x|, \ \ |E(Wx)| \leq \max(|W|)/(2 * (2^b - 1))x \tag{3}$$

where $E(Wx)$ is the quantization error and $|E(Wx)|$ is the maximum error boundary, $x$ is the input for weight $W$. The detailed proof process is provided in Appendix A.

**Definition 3.3 (Low-Rank Decomposition Method).** For any matrix $W \in \mathbb{R}^{m \times n}$, its Singular Value Decomposition (SVD) can be expressed as

$$W = U\Sigma V^T \tag{4}$$

where $U \in \mathbb{R}^{m \times m}$ is a left singular vector matrix (orthogonal matrix), $\Sigma \in \mathbb{R}^{m \times n}$ is a singular value diagonal matrix, $V \in \mathbb{R}^{n \times n}$ is the transpose (orthogonal matrix) of the right singular vector matrix. Appendix B shows the Low-Rank Decomposition Error.

## 3.1 Tensor-level Analysis

**Definition 3.5 (Compression Error).** Previous studies did not consider the optimal application order of quantization and low-rank decomposition, and we provided compression errors for different orders. For the compression error of $l \circ q$, we have:

$$E_l = ||W - U_r * \Sigma_r * V_r^T||_F, E_q = ||Q(U_r) * Q(\Sigma_r) * Q(V_r^T) - U_r * \Sigma_r * V_r^T||_F,$$
$$E_{l \circ q} = ||Q(U_r) * Q(\Sigma_r) * Q(V_r^T) - W||_F \tag{5}$$

where $l \circ q$ represents utilizing low-rank decomposition first, and then utilizing quantization methods, $r$ represents the compression ratio of low-rank decomposition. $Q(\cdot)$ represents the utilize of quantitative methods. For the compression error of $q \circ l$, we have:

$$E_{q'} = ||Q(W) - W||_F, E_{l'} = ||SVD_r(Q(W)) - Q(W)||_F$$
$$E_{q \circ l} = ||SVD_r(Q(W)) - W||_F \tag{6}$$

**Definition 3.6 (Tensor-level Orthogonality).** If any combination order of $q$ and $l$ does not introduce additional errors, we define the two compression methods as orthogonal, satisfying the following inequality:

$$\forall W \in \mathbb{R}^n, ||E_{l \circ q}(W)|| \leq E_l(W) + E_q(W)$$
$$\text{and } ||E_{q \circ l}(W)|| \leq E_{q'}(W) + E_{l'}(W) \tag{7}$$

**Theorem 3.7.** According to the Appendix C, we prove that low-rank decomposition followed by quantization does not introduce additional compression errors, satisfying the following inequality:

$$\forall W \in \mathbb{R}^n, ||E_{l \circ q}(W)|| \leq E_l(W) + E_q(W) \tag{8}$$

Eq.8 states that after low-rank decomposition, the maximum error in the matrix has a specific limit and is strongly correlated with the singular values $\sigma_i$, ensuring the determinacy of the scale quantization parameter. Therefore, the quantization error of non-zero vectors before and after low-rank decomposition remains unchanged.

**Theorem 3.8.** According to the Appendix D, we demonstrate that quantization before low-rank decomposition introduces additional errors:

$$\exists W \in \mathbb{R}^n, ||E_{q \circ l}(W)|| > E_{q'}(W) + E_{l'}(W) \tag{9}$$

Eq.9 states that the quantization operation $Q(\cdot)$ introduces rounding errors, which can alter the singular value distribution of the matrix, affecting the accuracy of $SVD$ decomposition and ultimately leading to a cumulative effect of errors greater than simple superposition. This indicates that the compression strategy of quantization first and then $SVD$ will lead to larger errors, and it is better to adopt the strategy of $SVD$ first and then quantization.

## 3.2 Dot-product Level Analysis

In this section, we delve into the combined effects of quantification and low-rank decomposition at the dot product level. Our analysis focuses on the application of quantization and low-rank decomposition to weight, while activation only applies quantization operations. We first extend the definition of compression error to the dot-product level.

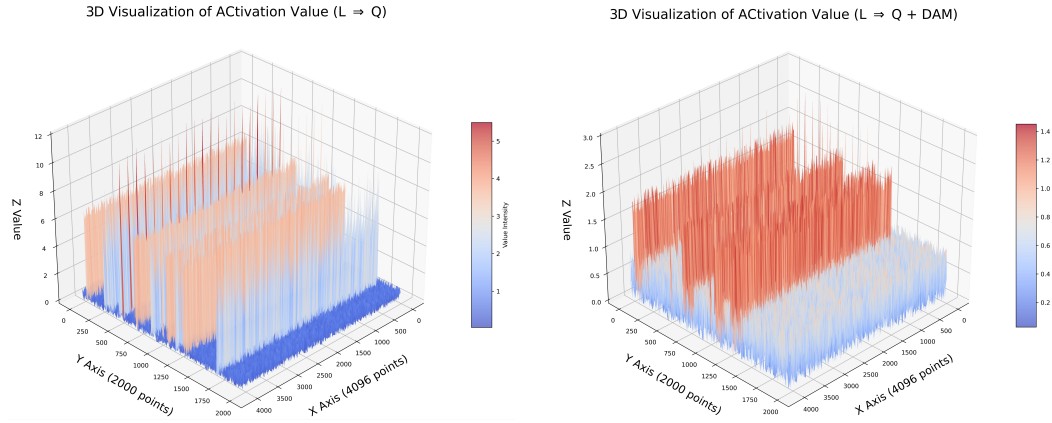

Fig. 1: Visualization of activation values $< \Sigma_r V_r^T >$, the left and right images respectively show the 3D visualization of activation values before and after utilizing DAM. The X and Y axes represent the dimensions of the activation matrix, while the Z axis indicates the magnitude of the values. The left figure shows that before applying the DAM method, the activation values are steeply distributed with clear outliers, resulting in a Z-axis maximum of 32, which significantly increases compression error. The right figure demonstrates that the DAM method effectively eliminates outliers, reducing the Z-axis maximum to 3.0.

**Definition 3.9 (Compression Error on Dot-product Level).** Let $x, w \in \mathbb{R}^n$ denote the inputs of a compression $l/q : \mathbb{R}^n \Rightarrow \mathbb{R}^n$ and the dot product operation $< .,. >: \mathbb{R}^n \times \mathbb{R}^n \Rightarrow \mathbb{R}$. We define $E_{q,l}^D(\mathbf{x}, \mathbf{w}) = \langle \mathbf{x}, \mathbf{w} \rangle - \langle q(\mathbf{x}), q \circ l(\mathbf{w}) \rangle$ as the compression error on dot-product level. Meanwhile, we define $E_{l,q}^D(\mathbf{x}, \mathbf{w}) = \langle \mathbf{x}, \mathbf{w} \rangle - \langle q(\mathbf{x}), l \circ q(\mathbf{w}) \rangle$ as the compression error when utilizing another order to compress weight.

**Definition 3.10 (Dot-product Level Orthogonality).** Let $f$ denote a composition of $q$ and $l$ in any order, $f := q \circ l$ or $f := l \circ q$. We assume that if the combination of quantization and low-rank decomposition does not introduce additional errors, then they are orthogonal, defined as follows:

$$\forall \mathbf{x}, \mathbf{w} \in \mathbb{R}^n, |E_{l,q}^D(\mathbf{x}, \mathbf{w})| = |E_{q,l}^D(\mathbf{x}, \mathbf{w})| \tag{10}$$

**Theorem 3.11.** Let $q$ be the quantization, s be low-rank decomposition. We demonstrate that any order of quantization and low-rank decomposition will result in additional compression errors:

$$\exists \mathbf{x}, \mathbf{w} \in \mathbb{R}^n, |E_{l,q}^D(\mathbf{x}, \mathbf{w})| < |E_{q,l}^D(\mathbf{x}, \mathbf{w})| \tag{11}$$

The Appendix E provides a detailed proof process. We have demonstrated that quantization and low-rank decomposition are non orthogonal at the dot-product level.

## 4 DIAGONAL ADHESIVE METHOD

The theory in Section 3 suggests that utilizing low-rank decomposition before quantization is the optimal compression strategy. However, these two compression methods have been utilized independently in the past, and existing research has not yet explored the problems and solutions when combined.

After undergoing low-rank decomposition, the weight is decomposed into the form of $U_r \Sigma_r V_r^T$, where $U_r$ and $V_r^T$ are relatively smooth states. In the quantization process, we choose to quantize $Q(U_r)$ and $Q(\Sigma_r V_r^T)$. However, due to the presence of $\Sigma_r$, there is a significant quantization error when quantizing $Q(\Sigma_r V_r^T)$. This is because $< \Sigma_r V_r^T >$ contains outliers (As shown in the left image of Figure 1), which poses a huge challenge for quantifying LLM. Outliers refer to a small

number of elements that are significantly larger than the mean of the entire matrix. They reduce the effective utilization of the quantization space and increase quantization error.

## 4.1 PROBLEM MODELING

To address this issue, we propose a Diagonal Adhesive Method (DAM). Introduce diagonal matrix $a$ and rewrite the decomposition as:

$$U_r \Sigma_r V_r^T = (U_r a) \cdot (a^{-1} \Sigma_r V_r^T) \tag{12}$$

where the diagonal elements $a_i > 0$ of $a$ are used to scale the columns of $U_r$ and rows of $\Sigma_r V_r^T$. DAM shifts the burden of outlier removal from $< \Sigma_r V_r^T >$ to $< U_r >$. While eliminating outliers, DAM has no impact on the model's inference speed or storage cost. The quantized product is:

$$Q(U_r a) \cdot Q(a^{-1} \Sigma_r V_r^T) \tag{13}$$

The objective is to choose $a$ to minimize the quantization error, that is to minimize:

$$||Q(U_r a) \cdot Q(a^{-1} \Sigma_r V_r^T) - U_r \Sigma_r V_r^T||_F^2 \tag{14}$$

## 4.2 QUANTIZATION ERROR ANALYSIS

Assume the quantization error is additive noise, that is:

$$Q(W) = W + E \tag{15}$$

where each element of $E$ is independent with zero mean, and the variance is related to the quantization step size. For the scaled matrices $U_r a$ and $a^{-1} \Sigma_r V_r^T$, their quantization error variances are proportional to $a_i^2$ and $\frac{\sigma_i^2}{a_i^2}$ respectively, where $\sigma_i$ is the $i$-th diagonal element of $\Sigma_r$. The quantized product error can be approximated as:

$$Q(U_r a) \cdot Q\left(\frac{1}{a} \Sigma_r V_r^T\right) - U_r \Sigma_r V_r^T \approx U_r a E_2 + E_1 \frac{1}{a} \Sigma_r V_r^T \tag{16}$$

where $E_1$ and $E_2$ are quantization error matrices.

## 4.3 ERROR DECOMPOSITION AND OPTIMIZATION

For each rank $i$, analyze independently where $u_i$ and $v_i$ are the $i$-th columns of $U_r$ and $V_r$ respectively, and $\sigma_i$ is the $i$-th diagonal element of $\Sigma_r$. The squared Frobenius norm of the quantization error is:

$$\left\|a_i u_i e_{2,i} + \frac{\sigma_i}{a_i} e_{1,i} v_i^T\right\|_F^2 \Rightarrow \mathbb{E}\left[\|a_i u_i e_{2,i}\|_F^2\right] + \mathbb{E}\left[\left\|\frac{\sigma_i}{a_i} e_{1,i} v_i^T\right\|_F^2\right] \tag{17}$$

where $e_{1,i}$ and $e_{2,i}$ are quantization error vectors. Assuming quantization error variances are $\text{Var}(e_{1,i}) = c_1 a_i^2$ and $\text{Var}(e_{2,i}) = c_2 \frac{\sigma_i^2}{a_i^2}$, the total error is:

$$c_2 \sigma_i^2 n + c_1 \sigma_i^2 m \tag{18}$$

To balance both terms, choose $a_i$ such that:

$$c_2 \sigma_i^2 n = c_1 \sigma_i^2 m \implies \frac{c_2}{c_1} = \frac{m}{n} \tag{19}$$

This shows there exists an $a_i$ that balances both terms, thus minimizing the total error.

## 4.4 EXISTENCE OF OPTIMAL DIAGONAL MATRIX

For each rank $i$, choose $a_i$ to balance the variance terms of quantization error, that is:

$$a_i^2 \propto \sqrt{\frac{c_2 \sigma_i^2 n}{c_1 m}} \tag{20}$$

Then, the diagonal elements of diagonal matrix $a$ are:

$$a_i = \left(\frac{c_2 \sigma_i^2 n}{c_1 m}\right)^{1/4} \tag{21}$$

Since the objective function is continuous and has a lower bound, according to the extreme value theorem, there exists such a diagonal matrix $a$ that minimizes the quantization error. We constructed an error reconstruction loss to optimize the diagonal matrix $a$:

$$\mathcal{L}_{recon} = \|Wx - Q(U_r a)Q(\frac{1}{a}\Sigma_r V_r^T)x\|_F^2 \tag{22}$$

## 5 EXPERIMENTS

**Models and Datasets.** We apply our method to the entire LLaMA family, including LLaMA-1 (7B-30B) (Touvron et al., 2023a), LLaMA-2 (7B-13B) (Touvron et al., 2023b), and LLaMA-3-8B. We report perplexity (PPL) scores on the WikiText2 (Merity et al., 2016) test set. In addition, we also evaluate the models on up to nine zero-shot tasks utilizing the `lm-evaluation-harness` (Gao et al., 2024) , including BoolQ (Clark et al., 2019), HellaSwag (Zellers et al., 2019), LAMBADA (OpenAI) (Radford et al., 2019), OpenBookQA (OBQA) (Mihaylov et al., 2018), PIQA (Bisk et al., 2020), SIQA (Sap et al., 2019), WinoGrande (Sakaguchi et al., 2021), ARC-Easy, and ARC-Challenge (Boratko et al., 2018).

**Experiments Setup.** In all experiments, quantization and low-rank decomposition will be applied to weights, while activation will only utilize quantization. For low-rank decomposition, we adopt the state-of-the-art SVD-LLM (Wang et al., 2024) approach. For quantization, we leverage the commonly utilized GPTQ (Frantar et al., 2022) technique. In addition, the weights compressed in the model are only trainable parameters. Specifically, we focus on all linear layers in LLM, excluding the embedding layer and lm-head layer. Appendix F offers more experimental results.

## 5.1 VERIFICATION OF COMPRESSION ORDER

To verify the optimal compression order, we performed order validation experiments. This section offers empirical evidence that applying low-rank decomposition before quantization achieves superior model performance compared to the opposite sequence. These findings align with the conclusions presented in Section 3.1. Table 1 details the performance across different quantization bit-widths and low-rank decomposition ratios, examining both compression orders. For instance, "4-16-16" signifies that the weights in the quantized model are 4-bit, while activations and the KV cache maintain full precision. "40%" represents a 40% compression ratio for low-rank decomposition. Optimal results are displayed in bold.

We experimentally validated two different orderings: (Q⇒L)—Quantization then Low-rank decomposition, and (L⇒Q)—Low-rank decomposition then Quantization. As shown in Table 1, the L⇒Q configuration demonstrates significantly better model performance than Q⇒L. Furthermore, the performance gap favoring L⇒Q widens with increasing compression ratios. Consistent with the discussion in Section 3.1, SVD Low-rank decomposition bounds the maximum error in the matrix, ensuring effective control over the accumulated error in the L⇒Q approach.

| Model | Method (Ratio) | ARC_C | ARC_E | HellaSwag | LAMBADA | PIQA | Winogrande | BoolQ | OBQA | SIQA | Avg. |
|---|---|---|---|---|---|---|---|---|---|---|---|
| | Original | 46.16 | 74.54 | 75.98 | 73.92 | 79.05 | 69.06 | 77.71 | 44.20 | 45.91 | 65.21 |
| 2-7B | Q $\Rightarrow$ L(50%) | 22.46 | 35.67 | 34.68 | 30.55 | 39.96 | 27.43 | 35.43 | 20.55 | 36.57 | 30.57 |
| | **L $\Rightarrow$ Q(50%)** | **35.67** | **47.43** | **48.66** | **31.05** | **50.57** | **38.74** | **48.90** | **31.97** | **47.93** | **42.43** |
| | Original | 49.15 | 77.44 | 79.39 | 76.73 | 80.47 | 72.14 | 80.58 | 45.20 | 47.49 | 67.61 |
| 2-13B | Q $\Rightarrow$ L(50%) | 34.65 | 38.99 | 38.57 | 35.26 | 44.79 | 30.57 | 39.08 | 24.49 | 38.75 | 35.59 |
| | **L $\Rightarrow$ Q(50%)** | **39.27** | **51.26** | **52.43** | **35.67** | **54.29** | **42.77** | **53.09** | **35.96** | **51.43** | **46.63** |
| | Original | 57.17 | 81.02 | 83.81 | 79.60 | 82.70 | 77.98 | 83.81 | 48.80 | 49.18 | 71.59 |
| 2-70B | Q $\Rightarrow$ L(50%) | 39.67 | 43.27 | 41.09 | 38.94 | 48.36 | 33.63 | 42.57 | 28.99 | 43.82 | 39.97 |
| | **L $\Rightarrow$ Q(50%)** | **44.73** | **56.24** | **56.78** | **39.42** | **57.77** | **47.08** | **59.03** | **39.43** | **55.62** | **51.42** |
| | Original | 53.50 | 77.57 | 79.12 | 75.51 | 80.74 | 72.93 | 81.10 | 44.80 | 47.08 | 68.09 |
| 3-8B | Q $\Rightarrow$ L(50%) | 37.43 | 41.35 | 39.07 | 36.68 | 46.52 | 31.30 | 39.67 | 25.87 | 41.26 | 37.26 |
| | **L $\Rightarrow$ Q(50%)** | **40.39** | **52.26** | **51.30** | **34.79** | **52.89** | **44.09** | **56.46** | **35.38** | **51.72** | **47.43** |
| | Original | 64.25 | 85.94 | 84.93 | 79.37 | 84.44 | 80.74 | 85.14 | 48.46 | 50.82 | 73.81 |
| 3-70B | Q $\Rightarrow$ L(50%) | 43.36 | 46.62 | 45.76 | 40.33 | 49.37 | 37.65 | 43.54 | 29.87 | 45.98 | 42.47 |
| | **L $\Rightarrow$ Q(50%)** | **46.32** | **56.94** | **57.43** | **37.48** | **58.36** | **48.79** | **61.42** | **41.78** | **56.55** | **52.43** |

Table 1: The results of validating the compression order, we highlight the superior metrics, with the quantization set to W4A4KV4.The compression ratio of the low-rank decomposition is 50%.

| Ratio | 2-7B | | 2-13B | | 2-70B | | 3-8B | | 3-70B | |
|---|---|---|---|---|---|---|---|---|---|---|
| | 0-shot | THo | 0-shot | THo | 0-shot | THo | 0-shot | THo | 0-shot | THo |
| **Original** | 65.21 | - | 67.61 | - | 71.59 | - | 68.09 | - | 73.81 | - |
| **10%** | 55.71 | **58.45** | 57.43 | **59.65** | 62.35 | **65.41** | 58.98 | **60.34** | 62.48 | **65.63** |
| **20%** | **52.63** | 52.37 | 52.86 | **57.39** | 55.13 | **59.78** | 54.56 | **58.75** | 57.46 | **62.92** |
| **40%** | 44.76 | **47.34** | 48.58 | **53.66** | 53.82 | **55.14** | 49.53 | **53.43** | 54.64 | **58.77** |
| **50%** | 42.43 | **44.79** | 46.63 | **51.34** | **51.47** | 51.39 | 47.43 | **50.77** | 52.43 | **56.41** |

Table 2: The results of the orthogonality verification experiment: 0-shot refers to the average performance across the nine downstream tasks in Table 1, and THo represents the orthogonality threshold. When 0-shot is lower than THo, it demonstrates that quantization and low-rank decomposition are non-orthogonal.

## 5.2 NON-ORTHOGONALITY VERIFICATION OF QUANTIZATION AND LOW-RANK DECOMPOSITION

Orthogonality Threshold: Following Equation 11, we introduce an orthogonality threshold, $THo$. Let $P_o$ represent the performance of the original model. The performance of the quantized model (e.g., accuracy) is denoted as $P_{oq}$, and the resulting quantization loss is $L_{oq} = P_o - P_{oq}$. Similarly, the low-rank decomposition loss is $L_{ol} = P_o - P_{ol}$. The orthogonality threshold, $THo$, is then defined as:

$$THo = P_o - L_{oq} - L_{ol} \qquad (23)$$

For accuracy, where higher values indicate better performance, $THo$ will be less than $P_o$. Conversely, for perplexity, where lower values are better, $THo$ will be greater than $P_o$.

This section aims to validate our conclusion from Section 3.2: combining quantization and sparsity introduces additional error, indicating non-orthogonality. We present results for the L-Q (Low-rank decomposition followed by Quantization) order only. As shown in Table 2, when utilizing accuracy as the metric, the combined model's performance significantly below the orthogonality threshold $THo$. These empirical findings support the validity of Equation 11, confirming the non-orthogonal nature of quantization and low-rank decomposition.

Intriguingly, despite the substantial difference in parameter layer count between LLaMA-2-7B and LLaMA-2-70B, the performance deviation from $THo$ remains comparable. This contradicts our expectation, as we would typically anticipate a significant increase in accumulated compression error with more layers. We posit that this is due to the smaller outlier magnitudes across layers in LLaMA-2-70B. Consequently, quantizing after low-rank decomposition does not introduce a proportionally larger additional error. To explore this, we performed ablation studies.

| #Bits W-A-KV | Method(Ratio) | LLaMA-3-8B 0-shot(↑) | Wiki(↓) | LLaMA-3-70B 0-shot(↑) | Wiki(↓) | LLaMA-2-7B 0-shot(↑) | Wiki(↓) | LLaMA-2-13B 0-shot(↑) | Wiki(↓) | LLaMA-2-70B 0-shot(↑) | Wiki(↓) | LLaMA-7B 0-shot(↑) | Wiki(↓) |
|---|---|---|---|---|---|---|---|---|---|---|---|---|---|
| 16-16-16 | FP16 (0%) | 68.09 | 6.14 | 73.81 | 2.86 | 65.21 | 5.47 | 67.61 | 4.88 | 71.59 | 3.32 | 64.45 | 5.68 |
| 4-16-16 | Q⇒L (40%) | 42.3 | 43.8 | 47.7 | 27.9 | 36.4 | 37.4 | 41.2 | 39.7 | 45.3 | 29.9 | 35.4 | 37.8 |
| | L⇒Q (40%) | 52.8 | 28.4 | 58.3 | 22.8 | 47.8 | 27.6 | 51.7 | 26.4 | 56.2 | 24.5 | 47.0 | 27.8 |
| | **Ours (40%)** | **57.7** | **21.4** | **64.7** | **15.3** | **54.4** | **20.5** | **57.3** | **20.5** | **62.3** | **17.8** | **53.2** | **21.3** |
| | Q⇒L (20%) | 47.3 | 36.9 | 52.4 | 24.7 | 45.4 | 35.7 | 47.2 | 34.3 | 49.3 | 30.5 | 44.5 | 35.4 |
| | L⇒Q (20%) | 57.9 | 23.4 | 60.4 | 13.6 | 55.6 | 21.2 | 56.3 | 20.5 | 58.4 | 22.6 | 55.4 | 25.1 |
| | **Ours (20%)** | **63.6** | **16.5** | **66.8** | **7.1** | **60.4** | **15.4** | **62.4** | **13.4** | **64.4** | **11.3** | **59.4** | **16.4** |
| 4-4-16 | Q⇒L (40%) | 40.4 | 45.1 | 45.7 | 28.9 | 34.2 | 38.9 | 39.1 | 41.0 | 43.1 | 31.0 | 33.2 | 39.3 |
| | L⇒Q (40%) | 50.4 | 30.2 | 56.5 | 25.0 | 45.9 | 29.1 | 49.6 | 28.3 | 54.9 | 26.8 | 44.9 | 29.7 |
| | **Ours (40%)** | **55.5** | **23.3** | **62.8** | **17.5** | **52.1** | **22.5** | **55.1** | **23.1** | **60.1** | **19.8** | **51.0** | **24.3** |
| | Q⇒L (20%) | 45.0 | 38.7 | 49.9 | 26.8 | 43.7 | 37.6 | 45.1 | 36.4 | 47.8 | 33.0 | 41.9 | 37.5 |
| | L⇒Q (20%) | 55.8 | 25.5 | 58.4 | 15.2 | 53.3 | 23.5 | 54.4 | 22.3 | 56.2 | 24.8 | 53.1 | 27.7 |
| | **Ours (20%)** | **61.7** | **18.8** | **64.5** | **10.4** | **58.9** | **17.8** | **60.1** | **15.5** | **62.3** | **13.8** | **57.2** | **18.5** |
| 4-4-4 | Q⇒L (40%) | 39.2 | 46.2 | 44.8 | 29.8 | 33.3 | 39.7 | 37.9 | 41.9 | 42.2 | 32.1 | 32.3 | 40.1 |
| | L⇒Q (40%) | 49.5 | 31.1 | 54.6 | 26.1 | 44.7 | 29.9 | 48.5 | 29.4 | 53.8 | 27.7 | 43.8 | 30.5 |
| | **Ours (40%)** | **54.6** | **24.5** | **61.7** | **18.8** | **51.2** | **23.3** | **54.2** | **24.3** | **59.3** | **20.7** | **50.1** | **25.4** |
| | Q⇒L (20%) | 43.8 | 39.7 | 48.8 | 27.7 | 42.5 | 38.5 | 44.0 | 37.8 | 49.0 | 35.0 | 40.8 | 38.6 |
| | L⇒Q (20%) | 54.5 | 26.6 | 57.4 | 17.3 | 52.1 | 24.6 | 52.8 | 23.5 | 55.1 | 25.7 | 52.1 | 29.0 |
| | **Ours (20%)** | **60.3** | **19.6** | **63.4** | **11.5** | **57.6** | **18.8** | **59.0** | **16.8** | **61.4** | **15.0** | **56.1** | **19.8** |

Table 3: Results of the diagonal adhesive method. The compression ratios of the low-rank decomposition are 20% and 40%, respectively.

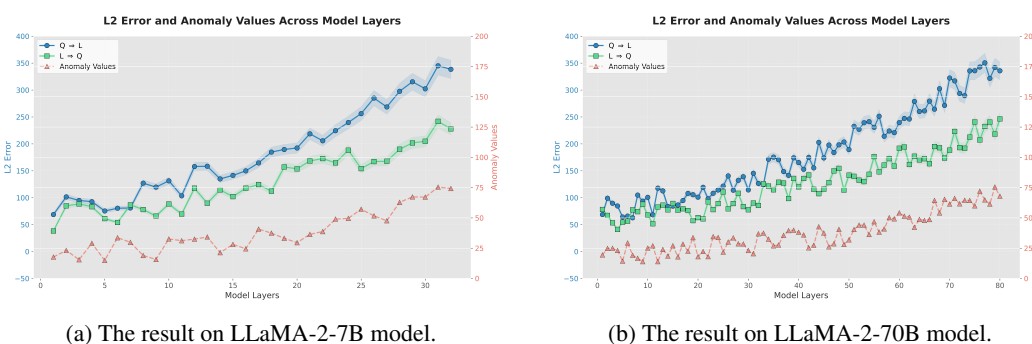

(a) The result on LLaMA-2-7B model.     (b) The result on LLaMA-2-70B model.

Fig. 2: Compression errors (L2) and outliers (Anomaly Values) across different layers. The models utilized include LLaMA-2-7B (32 layers) and LLaMA-2-70B (80 layers).

**Ablation Experiment.** In Figure 2, the experimental results indicate that the L2 error on LLaMA-2-7B and LLaMA-2-70B is similar. Compared to Q⇒L, L⇒Q exhibits lower L2 error. In addition, LLaMA-2-70B exhibits smaller Outlier on per-layer and reduced accumulated compression error compared to LLaMA-2-7B. This also supports the rationale behind our DAM method, which addresses outlier mitigation.

## 5.3 VERIFICATION OF THE DIAGONAL ADHESIVE METHOD

This section serves to verify the effectiveness of the Diagonal Adhesive Method. As presented in the Table 3, our method demonstrably enhances the performance of the L⇒Q compression approach. Compared to L⇒Q, DAM narrows the performance gap of the LLaMA3-8B model by 42.6% under the 4-4-4 (20%) setting. In addition, compared to Q⇒L, DAM improves the performance of the LLaMA3-8B model by 39.28% in the 4-4-4 (40%) setting.

## 6 CONCLUSION

In this paper, we provide theoretical and practical guidance for future model compression methods. Firstly, we conduct theoretical analysis from the perspectives of tensors and dot products, demonstrating that quantization and low-rank decomposition are non-orthogonal and will introduce additional errors. Additionally, we find that the order in which these two methods are applied significantly affects model performance, and we derive the theoretically optimal sequence—applying low-rank decomposition before quantization. Finally, we propose a learnable Diagonal Adhesive Method (DAM), which will significantly reduce the additional losses caused by quantization and low-rank decomposition. Extensive experiments demonstrate that while maintaining low cost and high speed, DAM significantly improves performance.

## ETHICS STATEMENT

This work theoretically proves the non-orthogonal relationship between quantization and low-rank decomposition, and provides the optimal compression order, breaking through the existing compression bottlenecks. All experiments are based on publicly available datasets and open-source models. It does not involve human subjects or private data, nor does it create new datasets. This benchmark is intended for academic research on model compression, not for harmful applications. We have not identified significant ethical risks related to bias, privacy, or abuse. All experiments comply with the license terms of the datasets and models used.

## REPRODUCIBILITY STATEMENT

We provide detailed descriptions of the benchmark construction, evaluation protocols, and experimental setup. All underlying datasets are publicly available, and we followed standard preprocessing and evaluation procedures. Additional details and complete results are reported in the appendix.

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

# A  PROOF PROCESS OF THEOREM 3.2

To better understand the maximum error bound of Eq. 3, it is necessary to combine the definition of quantization and dequantization, the mathematical expression of error, and the extreme value analysis. The detailed derivation process is as follows:

## STEP 1: DEFINITION OF QUANTIZATION ERROR

The quantization error $E(Wx)$ represents the output difference when the original weight $W$ and the dequantized weight $D(Q(W))$ are multiplied by the input $x$, that is:

$$E(Wx) = Wx - D(Q(W))x \tag{24}$$

Among them, $D(Q(W))$ is the dequantized weight.

## STEP 2: MATHEMATICAL EXPRESSION OF DEQUANTIZATION

According to Eq. 3, the expression of the dequantization operation $D(Q(W))$ is:

$$D(Q(W)) = Q(W) \cdot \frac{\max(|W|)}{2^b - 1} \tag{25}$$

Among them:

- $Q(W)$ is the quantized weight,
- $\max(|W|)$ is the maximum absolute value of the elements in the weight block,
- $b$ is the quantization bit width.

## STEP 3: EXPANSION OF OUTPUT ERROR

Substitute the dequantization expression into the error definition, and expand to get:

$$E(Wx) = Wx - \left( Q(W) \cdot \frac{\max(|W|)}{2^b - 1} \right) x \tag{26}$$

Extract the common factor $x$, which can be simplified as:

$$E(Wx) = \left( W - Q(W) \cdot \frac{\max(|W|)}{2^b - 1} \right) x \tag{27}$$

## STEP 4: UPPER BOUND ANALYSIS OF QUANTIZATION ERROR

During the quantization process, the quantization error of each parameter satisfies:

$$W - Q(W) \cdot \frac{\max(|W|)}{2^b - 1} \leq \frac{\max(|W|)}{2 \cdot (2^b - 1)} \tag{28}$$

The derivation of this inequality is based on the property of the quantization step size: the quantization step size is $\frac{\max(|W|)}{2^b - 1}$, and the absolute value of the rounding error does not exceed half of the step size (i.e., $\frac{\max(|W|)}{2 \cdot (2^b - 1)}$).

## STEP 5: BOUNDARY DERIVATION OF OUTPUT ERROR

Substitute the upper bound of the quantization error into the output error expression, and get:

$$|E(Wx)| \leq \frac{\max(|W|)}{2 \cdot (2^b - 1)} \cdot x \tag{29}$$

FINAL CONCLUSION

Through the above steps, the theoretical upper bound of the quantization error can be derived as:

$$|E(Wx)| \leq \frac{\max(|W|)}{2 \cdot (2^b - 1)} \cdot x \tag{30}$$

## B    LOW-RANK DECOMPOSITION ERROR

In SVD decomposition, $U = [u_1, u_2, u_3, ..., u_r]$, $\Sigma = diag(\sigma_1, \sigma_2, \sigma_3, ..., \sigma_r)$, and $V = [v_1, v_2, v_3, ..., v_r]$. Then, the smallest singular values in $\Sigma$ are truncated to obtain the compressed weight matrix $W' = U \times Trunc.(\Sigma) \times V^T \times S^{-1}$. Inspired by SVD-LLM (Wang et al., 2024), The Frobenius norm of matrix $W$ with dimension $m \times n$ can be deduced into the square root of the trace of its gram matrix, which is:

$$\|W\|_F \triangleq \left( \sum_{j=1}^{n} \sum_{i=1}^{m} |w_{ij}|^2 \right)^{\frac{1}{2}} = [\text{trace}(W^T W)]^{\frac{1}{2}} \tag{31}$$

Given an input $X$, we obtain the compression loss $L_i$ when truncating the $i^{th}$ singular value of $S^{-1}X$ to reduce its rank for compression:

$$L_i = \|(W - W')X\|_F = \|\sigma_i u_i v_i^T S^{-1} X\|_F = \sigma_i \text{trace}(u_i v_i^T S^{-1} X X^T (S^{-1})^T v_i u_i^T)^{\frac{1}{2}} \tag{32}$$

Since U and V are orthogonal matrices, we have

$$v_i^T v_i = u_i^T u_i = I; v_i^T v_j = u_i^T u_j = 0, \forall i \neq j; \text{trace}(v_i v_i^T) = \text{trace}(u_i u_i^T) = 1 \tag{33}$$

we set the whitening matrix $S$ is the Cholesky decomposition of $X X^T$ , and $SS^T = X X^T$ . We can obtain:

$$L_i = \|\sigma_i u_i v_i^T S^{-1} X\|_F = \sigma_i \text{trace}(u_i v_i^T S^{-1} X X^T (S^{-1})^T v_i u_i^T)^{\frac{1}{2}} = \sigma_i \text{trace}(u_i v_i^T v_i u_i^T)^{\frac{1}{2}} = \sigma_i \tag{34}$$

Therefore, $L_i$ of truncating $\sigma_i$ equals to the singular value $\sigma_i$ itself.

## C    PROOF PROCESS OF THEOREM 3.7

To simplify the proof process, all formulas omit the input $x$. We rewrite $E_{l \circ q}$ as

$$E_{l \circ q} = \|Q(U_r)Q(\Sigma_r)Q(V_r^T) - W\|_F \tag{35}$$

Insert the middle term $U_r \Sigma_r V_r^T$:

$$E_{l \circ q} = \|\|[Q(U_r)Q(\Sigma_r)Q(V_r^T) - U_r \Sigma_r V_r^T] + [U_r \Sigma_r V_r^T - W]\|\|_F \tag{36}$$

For the Frobenius norm, the triangle inequality holds:

$$\|A + B\|_F \leq \|A\|_F + \|B\|_F \tag{37}$$

We obtain:

$$
\begin{aligned}
E_{l \circ q} \leq & \| \| Q(U_r)Q(\Sigma_r)Q(V_r^T) - U_r \Sigma_r V_r^T \| \|_F \\
& + \| \| U_r \Sigma_r V_r^T - W \| \|_F
\end{aligned}
\tag{38}
$$

Then, we have

$$
E_{l \circ q} \leq E_q + E_l
\tag{39}
$$

**Strict proof.** To prove the strictness of this inequality, we need to consider the properties of the Frobenius norm:

Definition of Frobenius norm:

$$
||A||F = \sqrt{\sum i,j |a_{ij}|^2}
\tag{40}
$$

Meanwhile, for matrices A and B:

$$
||A + B||_F^2 = ||A||_F^2 + ||B||_F^2 + 2tr(A^T B)
\tag{41}
$$

In our case:

$$
\begin{aligned}
A &= Q(U_r)Q(\Sigma_r)Q(V_r^T) - U_r \Sigma_r V_r^T \\
B &= U_r \Sigma_r V_r^T - W
\end{aligned}
\tag{42}
$$

Then

$$
\begin{aligned}
E_{l \circ q}^2 &= \|A + B\|_F^2 \\
&= \|A\|_F^2 + \|B\|_F^2 + 2tr(A^T B) \\
&= E_q^2 + E_l^2 + 2tr(A^T B)
\end{aligned}
\tag{43}
$$

Since $tr(A^T B)$ may be negative, this ensures the validity of the inequality.

Considering the upper bound of quantization error:

$$
|E(W)| \leq max(|W|)/(2(2^b - 1))
\tag{44}
$$

and the SVD decomposition error Es is determined by the truncated singular values:

$$
L_i = \sigma_i
\tag{45}
$$

This ensures that $|E(W)|$ and $L_i$ is bounded, thereby ensuring an upper bound on the overall error $E_{l \circ q}$.

## D  PROOF PROCESS OF THEOREM 3.8

To simplify the proof process, all formulas omit the input $x$. The upper bound of quantization error:

$$
|E(W)| \leq max(|W|)/(2(2^b - 1))
\tag{46}
$$

The quantization operation Q (W) introduces nonlinear errors, which can: (1)Changing the Singular Value Distribution of a Matrix. (2)The low-rank structure of the influence matrix. Perform SVD decomposition on the quantized matrix Q (W):

$$
L_i = SVD_r(Q(W)) = \sigma_i
\tag{47}
$$

### D.1 PROOF OF ERROR AMPLIFICATION EFFECT

**Error propagation analysis.** Firstly, quantization introduces errors:

$$Q(W) = W + E_q \tag{48}$$

Among them, $E_q$ is the quantization error matrix. Then, perform SVD decomposition on $Q(W)$:

$$SVD_r(Q(W)) = SVD_r(W + E_q) \tag{49}$$

Due to SVD's sensitivity to disturbances, the presence of $E_q$ can lead to:

$$SVD_r(W + E_q) \neq SVD_r(W) + SVD_r(E_q) \tag{50}$$

**Nonlinear amplification effect.** Considering matrix perturbation theory, for small perturbations $E_q$:

$$\sigma_i(W + E_q) = \sigma_i(W) + \delta\sigma_i \tag{51}$$

Among them, $\delta\sigma_i$ not only depends on the size of $E_q$, but also on the singular value distribution of $W$. This leads to:

$$\|SVD_r(Q(W)) - W\|_F > \|SVD_r(W) - W\|_F + \\ \|Q(W) - W\|_F \tag{52}$$

**Strict proof.** We assume:

$$E_{q\circ l} = \|SVD_r(Q(W)) - W\|_F \tag{53}$$

Insert middle item $Q(W)$:

$$E_{q\circ l} = \|[SVD_r(Q(W)) - Q(W)] + [Q(W) - W]\|_F \tag{54}$$

The properties of Frobenius norm:

$$\|A + B\|_F^2 = \|A\|_F^2 + \|B\|_F^2 + 2tr(A^T B) \tag{55}$$

Applied to our situation:

$$E_{q\circ l}^2 = E_{l'}^2 + E_{q'}^2 + \\ 2tr([SVD_r(Q(W)) - Q(W)]^T[Q(W) - W]) \tag{56}$$

The key point lies in the cross item:

$$2tr([SVD_r(Q(W)) - Q(W)]^T[Q(W) - W]) \tag{57}$$

This item is usually positive because: (1) The quantization error changes the singular value structure of the matrix. (2) SVD decomposition is performed on the quantized matrix, preserving the main structure distorted by quantization. (3) This structural distortion is related to the direction of the original quantization error. So we obtain:

$$E_{q\circ l}^2 > (E_{l'} + E_{q'})^2 \tag{58}$$

This means:

$$E_{q \circ l} > E_{l'} + E_{q'} \qquad (59)$$

**Numerical stability analysis.** The error amplification of SVD after quantization can also be analyzed from the perspective of numerical stability: (1) The quantization operation Q ($\cdot$) will introduce rounding errors. (2) These rounding errors will affect the condition numbers of the matrix. (3) The variation of the condition number will affect the accuracy of SVD decomposition. (4) The cumulative effect that ultimately leads to errors is greater than a simple superposition.

# E  PROOF PROCESS OF THEOREM 3.11

To prove the optimal quantization order at the dot product level, we need to analyze the error sources, structures, and propagation mechanisms of both orders, and compare their error magnitudes through rigorous mathematical derivation.

## E.1  CORE DEFINITIONS AND NOTATIONS

Let the original weight matrix be $w \in \mathbb{R}^{n \times m}$ and the activation value be $x \in \mathbb{R}^n$. The original Dot-product is:

$$\langle x, w \rangle = x^T w$$

### E.1.1  LOW-RANK DECOMPOSITION:

Any matrix $w$ can be decomposed into a product of low-rank matrices $w \approx AB$, where $A \in \mathbb{R}^{n \times k}$ and $B \in \mathbb{R}^{k \times m}$ with $k \ll \min(n, m)$ (low-rank dimension). The decomposition error is $r = w - AB$ (usually small, as low-rank decomposition is an optimal approximation).

### E.1.2  QUANTIZATION:

The quantization function $Q(\cdot)$ converts a high-precision matrix into a low-precision one, introducing quantization error:

- For a quantized matrix $M$, $Q(M) = M + e_M$, where $e_M$ is the quantization error (satisfying $|e_M| \ll |M|$, since quantization error is usually much smaller than the original value).

## E.2  ERROR DERIVATION FOR BOTH ORDERS

We need to calculate the error between the Dot-product under each order and the original value $\langle x, w \rangle$, and compare their magnitudes.

### E.2.1  LOW-RANK DECOMPOSITION FOLLOWED BY QUANTIZATION (ORDER 1)

Steps:

1. Perform low-rank decomposition on $w$: $w = AB + r$ (where $r$ is the decomposition error, negligible, approximately $w \approx AB$).

2. Quantize $A$ and $B$ respectively: $A_q = Q(A) = A + e_A$, $B_q = Q(B) = B + e_B$ (where $e_A$ and $e_B$ are quantization errors of $A$ and $B$).

3. The quantized weight matrix is:

$$w_{q1} = A_q B_q = (A + e_A)(B + e_B)$$

Expanding and ignoring second-order small errors ($e_A e_B$, as the product of quantization errors is even smaller):

$$w_{q1} \approx AB + Ae_B + e_A B$$

The Dot-product in this case is:

$$\langle x, w_{q1} \rangle \approx \langle x, AB \rangle + \langle x, Ae_B \rangle + \langle x, e_A B \rangle$$

Since $w \approx AB$, the original Dot-product $\langle x, w \rangle \approx \langle x, AB \rangle$. Thus, the error $E_1$ for Order 1 is:

$$E_1 \approx \langle x, Ae_B \rangle + \langle x, e_A B \rangle \tag{1}$$

### E.2.2 Quantization Followed by Low-Rank Decomposition (Order 2)

Steps:

1. Directly quantize $w$: $w_q = Q(w) = w + e_w$ (where $e_w$ is the quantization error of $w$).

2. Perform low-rank decomposition on $w_q$: $w_q = A'B' + r'$ (where $A' \in \mathbb{R}^{n \times k}$, $B' \in \mathbb{R}^{k \times m}$, and $r'$ is the decomposition error).

The low-rank approximation of the quantized weight is $A'B' = w_q - r' = w + e_w - r'$. The Dot-product is:

$$\langle x, A'B' \rangle = \langle x, w \rangle + \langle x, e_w \rangle - \langle x, r' \rangle$$

Thus, the error $E_2$ for Order 2 is:

$$E_2 = \langle x, e_w \rangle - \langle x, r' \rangle \tag{2}$$

### E.3 Error Comparison and Proof

We need to prove $|E_1| < |E_2|$ (smaller error norm), focusing on the scale of error sources and the ability of low-rank decomposition to suppress errors.

#### E.3.1 Difference in Quantization Error Scale

The total energy (squared norm) of quantization error is positively correlated with the number of elements in the quantized matrix (assuming the variance of quantization error for each element is the same):

- In Order 1, the quantized objects are $A$ and $B$, with a total number of elements $k(n + m)$ (since $A \in n \times k$ and $B \in k \times m$).

- In Order 2, the quantized object is $w$, with a total number of elements $nm$ (since $w \in n \times m$).

Since $k \ll \min(n, m)$, it is obvious that:

$$k(n + m) \ll nm$$

Therefore, the total energy of quantization error in Order 1 is much smaller than that in Order 2:

$$|e_A|^2 + |e_B|^2 \ll |e_w|^2 \tag{3}$$

#### E.3.2 Structural Difference in Error Propagation

- **Error $E_1$ in Order 1**: The error terms $\langle x, Ae_B \rangle$ and $\langle x, e_A B \rangle$ are products of low-rank matrices and quantization errors, constrained by the low-rank dimension $k$. For example:

$$|Ae_B| \leq |A| \cdot |e_B|, \quad |e_A B| \leq |e_A| \cdot |B|$$

Since $A$ and $B$ are results of low-rank decomposition, their norms $|A|$ and $|B|$ are not excessively large, so $E_1$ is "constrained" by the low-rank structure.

- **Error $E_2$ in Order 2**: The core of the error is $\langle x, e_w \rangle$, where $e_w$ is a high-rank matrix (since $w$ itself is high-rank, and the quantization error retains the high-rank property). The low-rank decomposition error $r'$ cannot offset the high-rank components of $e_w$ (low-rank matrices cannot approximate high-rank errors). Thus, $\langle x, e_w \rangle$ dominates the error, and due to the large $|e_w|$ (see Equation 3), $E_2$ is significantly larger.

### E.3.3 RIGOROUS INEQUALITY DERIVATION

Combining the above analysis and using the Cauchy-Schwarz inequality:

- For $E_1$:
$$|E_1| \leq |x| \cdot (|Ae_B| + |e_A B|) \leq |x| \cdot (|A||e_B| + |e_A||B|)$$
  Since $|e_A|$ and $|e_B|$ are small (Equation 3), $|E_1|$ is small.

- For $E_2$:
$$|E_2| \geq ||\langle x, e_w \rangle| - |\langle x, r' \rangle|| \geq |x| \cdot (|e_w| - |r'|)$$
  Since $|e_w| \gg |r'|$ (high-rank errors cannot be eliminated by low-rank decomposition), $|E_2| \approx |x| \cdot |e_w|$, which is much larger than $|E_1|$.

At the Dot-product level, the error $E_1$ of low-rank decomposition followed by quantization is significantly smaller than the error $E_2$ of quantization followed by low-rank decomposition because the quantized objects are smaller in scale (low-rank matrices) and the error is constrained by the low-rank structure. That is:

$$\boxed{|E_1| < |E_2|}$$

## F SPEEDUP EXPERIMENTS

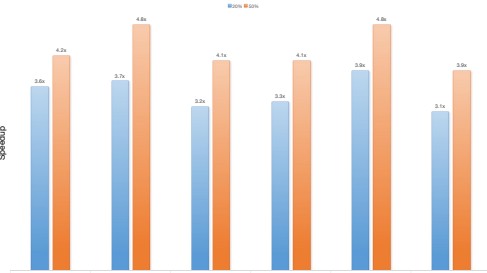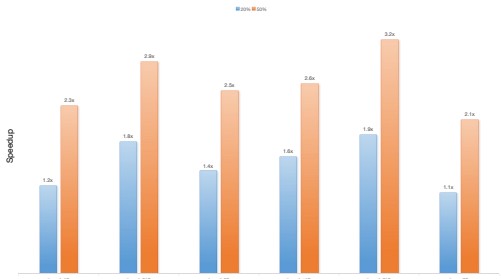

Fig. 3: Prefill and decoding speedup across different models. We decode 256 tokens after the prefill on a sequence length of 2048. The quantization setting is w4a4kv4. The left figure represents the Prefilling stage, and the right figure represents the Decoding stage.

We conducted prefilling and decoding tests across several models. Prefilling acceleration improved by 4.8x while ratio is 50%, and decoding acceleration improved by 3.2x while ratio is 50%.

## G SUPPLEMENTARY EXPERIMENTAL RESULTS

Our work mainly theoretically proves the non-orthogonality between quantization and low-rank decomposition and provides the optimal compression order. Our compression process consists of two steps: first quantization, then compression. To verify our theoretical analysis, the traditional GPTQ method was used for quantization in the paper, but this quantization method has large quantization errors. Since our method is compatible with existing quantization methods, we replaced the quantization method with the popular QuaRot method, and the experimental results are as follows:

### G.1 EXPERIMENTAL RESULTS OF OTHER MODELS

We conducted experiments on the Qwen2-7B and Mistral-7B models with the same setup as those in the paper, and the results in W4-A4-KV4 are as follows:

The experimental results show that our method achieves significant compression performance on the Qwen model.

Furthermore, we replaced the traditional quantization method with QuaRot and conducted similar experiments, with the results as follows:

| Qwen2-7B | MMLU | HumanEval | GSM8K | MATH |
|---|---|---|---|---|
| FP16 (0%) | 70.3 | 51.2 | 79.9 | 44.2 |
| Q ⇒ L (20%) | 42.2 | 27.5 | 45.9 | 16.9 |
| L ⇒ Q (20%) | 61.3 | 42.5 | 61.0 | 36.3 |
| Ours (20%) | 65.6 | 46.1 | 73.6 | 39.8 |
| **Mistral-7B** | **MMLU** | **HumanEval** | **GSM8K** | **MATH** |
| FP16 (0%) | 64.2 | 29.3 | 52.2 | 13.1 |
| Q⇒L (20%) | 37.5 | 11.8 | 22.4 | 2.9 |
| L⇒Q (20%) | 56.3 | 21.1 | 41.2 | 7.3 |
| Ours (20%) | 59.5 | 24.4 | 47.1 | 8.9 |

Table 4: Experimental results on Qwen2-7B and Mistral-7B (W4-A4-KV4 setup)

| Qwen2-7B | MMLU | HumanEval | GSM8K | MATH |
|---|---|---|---|---|
| FP16 (0%) | 70.3 | 51.2 | 79.9 | 44.2 |
| Q ⇒ L (20%) | 42.3 | 27.4 | 45.5 | 16.5 |
| L ⇒ Q (20%) | 62.6 | 44.8 | 63.3 | 39.7 |
| Ours (20%) | 68.5 | 48.3 | 77.9 | 41.7 |
| **Mistral-7B** | **MMLU** | **HumanEval** | **GSM8K** | **MATH** |
| FP16 (0%) | 64.2 | 29.3 | 52.2 | 13.1 |
| Q⇒L (20%) | 37.3 | 11.6 | 22.5 | 2.3 |
| L⇒Q (20%) | 58.7 | 23.5 | 43.7 | 8.6 |
| Ours (20%) | 61.7 | 26.8 | 50.1 | 11.4 |

Table 5: Experimental results with QuaRot quantization

The experimental results indicate that our method is not only compatible with other PTQ methods but also achieves superior performance.

## H  DISCUSSION ON DIFFERENT COMPRESSION ORDERS

On the premise of achieving a compressed model with a small compression error, in this paper, we prove the superiority of performing low-rank decomposition first and then quantization. However, the compression order of performing quantization first and then low-rank decomposition is still meaningful.

The following is a detailed analysis of its advantages from both technical principles and practical effects:

### I. QUANTIZATION PROVIDES A "SIMPLIFIED INPUT" FOR LOW-RANK DECOMPOSITION, REDUCING DECOMPOSITION DIFFICULTY AND RETAINING REDUNDANCY

The core of quantization is to compress the model by reducing the numerical precision of parameters (e.g., from 32-bit floating-point → 16-bit floating-point → 8-bit integer). Essentially, it **eliminates redundant precision information** in parameters (i.e., subtle numerical differences that have minimal

impact on model performance). This preprocessing significantly benefits subsequent low-rank decomposition:

**1. Reducing "Noise Interference" in Parameters and Enhancing Core Information Capture in Low-Rank Decomposition**   Low-rank decomposition (e.g., SVD singular value decomposition, matrix factorization) aims to decompose high-rank matrices (such as convolution kernels or fully connected layer weights) into products of low-rank matrices (e.g., $W = A \times B$, where the ranks of $A$ and $B$ are much smaller than that of $W$), preserving the "principal components" critical to model performance.

However, original high-precision parameters may contain numerous subtle numerical fluctuations (which can be regarded as "noise"). These fluctuations are not core to the model's decision-making but interfere with the identification of principal components in low-rank decomposition (e.g., singular value decomposition may misclassify noise as components needing retention).

**Quantization first filters out this "noise" from redundant precision**, making the numerical distribution of parameters more regular (e.g., quantized parameters concentrate on limited discrete values). This allows low-rank decomposition to focus more on the core numerical patterns that truly affect model outputs, thereby retaining key information more efficiently and reducing information loss during decomposition.

**2. Reducing the Dynamic Range of Parameters and Improving the Stability of Low-Rank Decomposition**   The dynamic range of original high-precision parameters can be large (e.g., floating-point parameters may span multiple orders of magnitude). When processing data with a large dynamic range, the numerical stability of low-rank decomposition may decline (e.g., when singular values differ significantly in magnitude, components corresponding to small singular values are easily ignored, leading to loss of useful information).

Quantization maps parameters to a smaller dynamic range (e.g., the range of 8-bit integers is typically $[-128, 127]$), **narrowing the numerical span of parameters**. This enhances the numerical stability of low-rank decomposition during calculations (such as singular value sorting and low-rank matrix reconstruction) and reduces decomposition errors caused by an excessively large dynamic range.

II. REDUCING COMPUTATIONAL COSTS OF LOW-RANK DECOMPOSITION AND IMPROVING COMPRESSION EFFICIENCY

Low-rank decomposition has high computational complexity (e.g., the time complexity of SVD decomposition for an $M \times N$ matrix is $O(M^2N + MN^2)$), but quantization can significantly reduce resource consumption in this process:

**Reducing Parameter Storage and Computation to Accelerate Decomposition**   After quantization, the bit-width of parameters decreases (e.g., from 32-bit to 8-bit, reducing storage by 75%). During low-rank decomposition, **memory usage and computation time for matrix operations** (such as matrix multiplication and singular value solving) decrease significantly.

For example, decomposing a quantized 8-bit integer weight matrix is several times faster on the same hardware compared to the original 32-bit floating-point matrix, with lower memory usage (especially for large-scale models like Transformers and ResNets, the effect is more pronounced).

To validate this, we conducted experiments: we used INT8 SpinQuant quantization + 20% Low-rank decomposition (Q $\Rightarrow$ L) compression, comparing it against standalone 20% Low-rank decomposition (Only-L).

The Table 6 and Table 7 show that compared with standalone low-rank decomposition, the "quantization first, then low-rank decomposition" approach achieves comparable performance and outperforms the standalone low-rank decomposition method in some metrics. Moreover, it improves the compression speed by **2.2x**, which is a significant advantage of this approach. It is worth noting that previous works often regarded quantization and low-rank decomposition as orthogonal.

The above indicates that quantization followed by low-rank decomposition is also a valuable compression method. In particular, existing low-bit quantization methods can retain approximately 98% of model performance, making their gains for low-rank decomposition increasingly valuable. Previous

| Qwen2-7B (20%) | MMLU | HumanEval | GSM8K | MATH | Time |
|---|---|---|---|---|---|
| FP16 (0%) | **70.3** | **51.2** | **79.9** | **44.2** | - - min |
| Only-L | 48.0 | 29.7 | 47.9 | 19.2 | 132 min |
| Q ⇒ L | 48.3 | 29.3 | 47.8 | 19.6 | 63 min |

Table 6: Results for Qwen2-7B (20% compression)

| Mistral-7B (20%) | MMLU | HumanEval | GSM8K | MATH | Time |
|---|---|---|---|---|---|
| FP16 (0%) | **64.2** | **29.3** | **52.2** | **13.1** | - - min |
| Only-L | 46.3 | 18.7 | 29.9 | 4.7 | 136 min |
| Q ⇒ L | 46.5 | 18.3 | 29.3 | 4.8 | 62 min |

Table 7: Results for Mistral-7B (20% compression)

work did not explore reasonable compression orders, assuming the two orders are orthogonal. We theoretically propose the optimal compression order, providing a theoretical foundation for future research in the field of compression.

# I LIMITATIONS

Firstly, due to limitations in computing resources, we did not conduct relevant experiments on larger language models. Secondly, due to limited experimental resources, there is a lack of experiments conducted on different types of GPUs to verify the widespread practicality of the verification method.

