# OpenReview forum: "Effective Interaction between Quantization and Low-Rank Decomposition based on LLMs"
_ICLR.cc/2026/Conference — ICLR 2026 Conference Withdrawn Submission_

### Official Review · Reviewer_Xku7 · 2025-10-26

**Soundness:** 1
**Presentation:** 1
**Contribution:** 1
**Rating:** 0
**Confidence:** 5

**Summary:**

This paper aims to maximize the accuracy of the compressed models by jointly applying low-rank approximation and quantization. This is a critical problem for mitigating the issue of quantization, which often exhibits severe performance degradation at 3 bits or lower. The paper proposes a new technique based on a theoretical analysis in this area.

However, contrary to the paper's goal of proposing a technique through a theoretical approach, the paper contains severe theoretical errors. The major issues are as follows:

* The paper's experiments deal with activation quantization, but Definition 3.1 only covers weight quantization, which falls outside the scope of the definition's applicability.

* Definition 3.2 considers the activation when defining quantization error, yet Definition 3.5 does not. Furthermore, an error term for low-rank decomposition is used without being properly defined.

* The equations in the Appendix inherently contain numerous typos. Additionally, line 787 states that $\text{tr}(A^T B)$ "may be" negative, which is unacceptable phrasing to be included in a rigorous mathematical proof.

Based on these issues, this reviewer finds the paper's theoretical content unreliable, and the experimental results built upon this theoretical analysis are also difficult to trust. I personally suspect the paper was entirely written using an LLM. Therefore, this reviewer recommends a **strong rejection**.

**Strengths:**

No strength,

**Weaknesses:**

See the summary.

**Questions:**

No questions.

**Details Of Ethics Concerns:**

This article has significantly unreliable theoretical statements, which should not be reused in other papers. Authors who submit their own paper to OpenReview have the responsibility to check their content strictly before submission,

---

### Official Review · Reviewer_NNrR · 2025-10-30

**Soundness:** 1
**Presentation:** 3
**Contribution:** 2
**Rating:** 2
**Confidence:** 4

**Summary:**

This paper investigates the interaction between quantization and low-rank decomposition for large language model compression. The authors argue that these two techniques are not orthogonal and that the optimal order is Low-rank decomposition → Quantization (L→Q) rather than Quantization → Low-rank decomposition (Q→L). They further propose a Diagonal Adhesive Method (DAM) to mitigate quantization errors by rank-wise scaling using a learnable diagonal matrix.
While the paper is clearly written and the empirical results are extensive, the central problem formulation contains a fundamental flaw:
Q→L (quantization followed by low-rank decomposition) is not a valid or meaningful pipeline in either theory or practice.

**Strengths:**

- The paper provides a unified framework to discuss how quantization interacts with low-rank decomposition.
- DAM itself is a simple yet practical idea that empirically improves robustness.
- Experimental results on multiple LLaMA variants are detailed and reproducible.

**Weaknesses:**

- Invalid problem setup (Q→L is conceptually meaningless).
Quantization converts floating-point weights into integer form to remove float-domain computation. Performing SVD on quantized weights necessarily requires dequantization, which reintroduces floats—thus invalidating the quantization step itself. The paper treats “Quantization → Low-rank decomposition” (Q→L) as a legitimate alternative order, but this operation cannot exist in an actual quantized inference pipeline. As a result, comparing L→Q and Q→L is not a theoretically symmetric problem, but rather a comparison between a valid and an invalid setting.

- Overstated novelty and misleading framing.
The authors repeatedly claim to be the first to prove that quantization and low-rank decomposition are non-orthogonal. However, L→Q and Q→L result in different values as Q→L is an invalid setting. Therefore, proving non-orthogonality between the two operations is not a discovery rather a restatement of basic linear algebra.
Also, no prior work assumed they were orthogonal in the first place. The paper’s framing suggests resolving a nonexistent controversy.

- Incremental idea.
The diagonal rescaling (DAM) is an incremental extension of channel-wise or block-wise scaling (e.g., SmoothQuant (ICML’23), Basis-sharing (ICLR’26)). While useful, it does not constitute a fundamentally new principle.

**Questions:**

- Can Q→L be implemented in a truly quantized (int8 or int4) inference pipeline?
- If Q→L is infeasible in practice, what conceptual meaning would your analysis of “non-orthogonality” still hold?
- How does your diagonal rescaling differ fundamentally from SmoothQuant’s per-channel scaling or basis-sharing normalization?
- Given that the paper derives the existence of an optimal diagonal matrix, what is the rationale for learning it through gradient updates?

---

### Official Review · Reviewer_Rfkt · 2025-11-01

**Soundness:** 2
**Presentation:** 2
**Contribution:** 2
**Rating:** 4
**Confidence:** 2

**Summary:**

This paper investigates the interaction between quantization and low-rank decomposition for compressing large language models (LLMs). While prior work often assumes these two methods are orthogonal (i.e., their errors are independent), the authors provide theoretical proofs and empirical evidence that they are in fact non-orthogonal, meaning their combination introduces additional errors. The paper further shows that the order of applying the methods matters, with low-rank decomposition before quantization being superior. To mitigate performance degradation, the authors propose a Diagonal Adhesive Method (DAM), which reduces activation outliers and improves quantization robustness. Experiments on the LLaMA family (7B–70B) across multiple benchmarks demonstrate that DAM significantly improves performance under combined compression.

**Strengths:**

1. This paper provides the first formal proof that quantization and low-rank decomposition are non-orthogonal.
2. This paper proposes DAM, a lightweight diagonal scaling technique that effectively reduces quantization error caused by outliers.

**Weaknesses:**

1. Generality beyond LLaMA: Experiments are restricted to LLaMA models; it is unclear whether the findings generalize to other architectures (e.g., Qwen3).

**Questions:**

1. Have you tested DAM on non-LLaMA architectures, or do you expect the same non-orthogonality and order sensitivity to hold universally?

---

### Official Review · Reviewer_yMcP · 2025-11-01

**Soundness:** 1
**Presentation:** 2
**Contribution:** 1
**Rating:** 0
**Confidence:** 3

**Summary:**

The authors aim to derive theoretical insights on orthogonality and the proper order of factorization and quantization of large language models (LLMs). They further introduce the diagonal adhesive method (DAM) to account for outliers.

**Strengths:**

- The paper attempts to provide a theoretical justification for previously observed effects regarding joint factorization and quantization.
- The application of FWSVD/ASVD diagonal scaling is adapted to a slightly different task of joint quantization and factorization.

**Weaknesses:**

- The paper does not consider existing methods that employ both quantization and low-rank decomposition, such as QLoRA and Quantization-Aware Factorization.
- The conclusions about the order of factorization and quantization and their orthogonality are largely self-evident. While the mathematical proofs may interest some, the broader impact and usefulness of these results are questionable.
- The proposed DAM approach is too conceptually similar to ASVD and FWSVD
- There are several inconsistencies throughout the text that reduce readability. More details are listed in the “Questions” section.
- The paper does not provide comparisons with any other compression methods, which is a huge drawback.

**Questions:**

- There should be a clear indication that symmetric quantization is used in Equation 1; otherwise, a zero-point term should be included.
- There is an inconsistency between Equations 1 and 2 regarding the step size. Please define the step size properly.
- Equation 3 should use proper vector/tensor norm notations rather than the absolute value definition.
- Existing papers that combine quantization and factorization, such as QLoRA or Quantization-Aware Factorization, are not cited or discussed.
- The procedure in Equation 6 appears questionable: taking the SVD of quantized weights may be meaningless, as the factors will no longer align with the quantization grid. It is unclear why this comparison is necessary.

---

### Note · Authors · 2025-12-26

I have read and agree with the venue's withdrawal policy on behalf of myself and my co-authors.